# *Stagonosporopsis pogostemonis*: A Novel Ascomycete Fungus Causing Leaf Spot and Stem Blight on *Pogostemon cablin* (Lamiaceae) in South China

**DOI:** 10.3390/pathogens10091093

**Published:** 2021-08-27

**Authors:** Zhang-Yong Dong, Ying-Hua Huang, Ishara S. Manawasinghe, Dhanushka N. Wanasinghe, Jia-Wei Liu, Yong-Xin Shu, Min-Ping Zhao, Mei-Mei Xiang, Mei Luo

**Affiliations:** 1Innovative Institute for Plant Health, Zhongkai University of Agriculture and Engineering, Guangzhou 510225, China; dongzhangyong@zhku.edu.cn (Z.-Y.D.); yinghua@zhku.edu.cn (Y.-H.H.); 18814383032@163.com (J.-W.L.); shuyx799@163.com (Y.-X.S.); zmp0609@163.com (M.-P.Z.); mm_xiang@163.com (M.-M.X.); 2Center for Mountain Futures, Kunming Institute of Botany, Chinese Academy of Sciences, Honghe 654400, China; dnadeeshan@gmail.com

**Keywords:** Didymellaceae, phoma-like, pathogenicity, phylogeny

## Abstract

*Pogostemon cablin* is one of the well-known Southern Chinese medicinal plants with detoxification, anti-bacterial, anti-inflammatory, and other pharmacological functions. Identification and characterization of phytopathogens on *P. cablin* are of great significance for the prevention and control of diseases. From spring to summer of 2019 and 2020, a leaf spot disease on *Pogostemon cablin* was observed in Guangdong Province, South China. The pathogen was isolated and identified based on both morphological and DNA molecular approaches. The molecular identification was conducted using multi-gene sequence analysis of large subunit (LSU), the nuclear ribosomal internal transcribed spacer (ITS), beta-tubulin (*β-tubulin*), and RNA polymerase II (*rpb2*) genes. The causal organism was identified as *Stagonosporopsis pogostemonis*, a novel fungal species. Pathogenicity of *Stagonosporopsis pogostemonis* on *P. cablin* was fulfilled via confining the Koch’s postulates, causing leaf spots and stem blight disease. This is the first report of leaf spot diseases on *P. cablin* caused by *Stagonosporopsis* species worldwide.

## 1. Introduction

*Pogostemon cablin* is a perennial aromatic herb, generally known as “Patchouli” or “Guanghuoxiang” [1]. This is wildly used as a herbal and a raw material of aromatics which have high economic demand for its essential oil [2,3]. *Pogostemon cablin* adapts to hot and humid climatic conditions and is mostly cultivated in the Philippines and some other tropical regions, including China. This plant is wildly grown in Southern China, including Guangdong, Guangxi, Hainan, and Fujian regions, as well as Taiwan [4]. As a well-known medicinal plant in traditional Chinese medicine, *Pogostemon cablin* is recognized for its detoxification, and other pharmacological features [5]. It contains a variety of natural antibacterial compounds, and has antibacterial and anti-inflammatory effects [6]. However, *P. cablin* is often affected by various diseases during its growth and development, that affect medicinal value [7]. Bacterial wilt caused by *Ralstonia solanacearum* [8], root-knot disease caused by *Meloidogyne incognita* [9,10], some viral diseases [11], and the fungal disease by *Corynespora cassiicola* causing leaf spot [12] are reported.

The species of *Stagonosporopsis* (Didymellaceae, Pleosporales) [13,14] can cause devastating diseases on a wide range of economically important plants, those in farmlands, forests, grasslands, and other natural ecosystems [13]. Moreover, they can result in severe yield loss of agricultural crops [15]. *Stagonosporopsis* species have been reported as severe pathogens of some crops and ornamentals in several countries, including Australia [16], China [17], France [18], India [19], Italy [20], Turkey [21], and the United States [22,23].

During 2019–2020, a new leaf disease was observed in the *Pogostemon cablin* orchards in Zhanjiang city of Guangdong Province, China. Samples were collected and the causal pathogen was isolated. The objectives of this study were to identify the fungal taxa causing leaf spots and stem blight disease of *P. cablin* in China by a combination of morphological characterization and phylogenetic analyses and confirm the pathogenicity by fulfilling Koch’s postulates.

## 2. Results

### 2.1. Field Symptoms

The disease incidence was 15% under high temperature, above 25–28 °C, and high humidity. Disease symptoms initially start with foliar marginal brown discoloration, then spreading towards the inside leaf lamina. Moreover, the disease spreads to cover completely the leaf, turning it dark brown and dry (Figure 1).

### 2.2. Morphological and Molecular Characterization

In total, 10 samples were collected from the field. The causal organism was isolated by tissue isolation method and pure cultures were obtained. Almost all the cultures obtained on PDA were morphologically similar, therefore two isolates were obtained for further analyses. For the molecular characterization, genomic DNA was extracted and four gene regions; LSU, ITS, *rpb2*, and *β-tubulin* were sequenced (Table 1). Phylogenetic analyses were done using maximum likelihood (ML), maximum parsimony (MP), and Bayesian analyses. Combined sequence data set of LSU, ITS, *rpb2*, and *β-tubulin* comprised of these two *Stagonosporopsis* isolates from this study and 76 reference sequences. The tree was rooted with *Allophoma piperis*. The tree topology of the ML analyses was similar to the MP and Bayesian analysis. Therefore, the best scoring RAxML tree with a final likelihood value of −10,074.586469 is presented (Figure 2). The matrix had one partition and 493 distinct alignment patterns, with 10.95% of undetermined characters or gaps. Estimated base frequencies were as follows: A = 0.243855, C = 0.237878, G = 0.274045, and T = 0.244222; substitution rates AC = 2.129010, AG = 4.916610, AT = 1.711303, CG = 0.847171, CT = 13.823203, and GT = 1.00; gamma distribution shape parameter α = 0.096067. The dataset consisted of 2680 characters with 2256 constant characters and 364 (13.6%) parsimony-informative and 60 parsimony-uninformative characters. The maximum number of trees generated was 5000, and the most parsimonious trees had a tree length of 1148 (CI = 0.496, RI = 0.819, RC = 0.406, and HI = 0.504). Two isolates obtained in this study were clustered together in an individual clade with 100% maximum likelihood (ML), 100% maximum parsimony bootstrap values (MP), and 1.0 Bayesian posterior probabilities values (BYPP) (Figure 2). Therefore, both morphology and pairwise nucleotide differences among strains isolated in this study the type species *S. hortensis* were compared. Based on molecular and morphological evidences, isolates obtained in this study (ZHKUCC 21-0001 and ZHKUCC 21-0002) were identified as *Stagonosporopsis pogostemonis* a novel species. The species description is given below:

*Stagonosporopsis pogostemonis* M. Luo, Y.H. Huang, & Manawas., sp. nov.;

Index Fungorum number: IF558590, Faces of fungi: FoF 09997, MycoBank accession number: MB840964 (Figure 3);

*Etymology*–In reference to the host genus name *Pogostemon*;

*Holotype*–ZHKUCC 21-0001;

Pathogenic on *Pogostemon cablin* leaves. Sexual morph: Not observed. Asexual morph: *Conidiomata* solitary, scattered, mostly covered with dense vegetative hyphae, subglobose or lageniform, dark brown to black, thick-walled, pycnidial. *Pycnidia* 34–161 µm × 32–98 µm ($\overset{-}{x}$ = 58 ± 27 µm × 45 ± 16 µm, n = 50), subglobose, arranged irregularly in a disc, with conidium and a circular or longitudinal ostiole. *Pycnidial wall* pseudoparenchymatous, composed of several layers of angularis cells. *Conidiogenous cells* not observed. *Conidia* 3–5 μm × 1–3 μm ($\overset{-}{x}$ = 4 ± 0.4 μm × 2 ± 0.3 μm, n = 50), oblong, cylindrical to ellipsoidal, with rounded both ends, smooth-walled, aseptate, with two polar guttules;

Culture characteristics: Colonies on PDA, MEA, and OA had covered the entire surface of the plate (8.5 cm in diameter) after seven days at 25 °C, growth rate of 10–16 mm/d. Colonies on PDA, margin regular, cottony, white toward the periphery, brownish-grey in the colony centre. Reverse white, becoming tawny then dark brown–black from the center. The colony characters on MEA were similar as on PDA. Colonies on OA grey and light grey. Reverse white, becoming grey then dark grey–black from the center;

Material examined: China, Guangdong Province, Zhanjiang, isolated from diseased leaves of *Pogostemon cablin*, April 2020, Y.H. Huang and Y.X. Shu, (dried culture ZHKU 21-0001, holotype and ZHKU 21-0002, paratype); ex-type culture ZHKUCC 21-0001 and ex-paratype culture ZHKUCC 21-0002; and

Habitat and host: On diseased leaves of *Pogostemon cablin*;

Known distribution: China (Zhanjiang, Guangdong province).

Notes: In the multigene phylogenetic tree constructed using LSU, ITS, *rpb2*, and *β-tubulin*, our new isolates (ZHKUCC 21-0001 and ZHKUCC 21-0002) constituted a monophyletic clade with 100% maximum likelihood, 100% maximum parsimony bootstrap, and 1.00 Bayesian posterior probability values. Thus, based on the phylogenetic species concept, we introduce this species as a new *Stagonosporopsis* species causing disease in *P. cablin.*
pathogens-10-01093-t001_Table 1Table 1Strains used for the phylogenetic analyses in this study and their GenBank accession Numbers.NameStrain Number ^1^Substrate (Including Host)GenBank Accession Numbers ^2^*rpb2**tub2*LSUITS*Stagonosporopsis Actaeae* *CBS 106.96; PD 94/1318*Actaea spicata*KT389672GU237671GU238166GU237734*S. actaeae*CBS 114303; UPSC 2962*Actaea spicata*—KT389847KT389760KT389544*S. actaeae*CBS 105.96; PD 74/230*Cimicifuga simplex*MT018018GU237670GU238165GU237733*S. ailanthicola* *MFLUCC 16-1439*Ailanthus altissima*KY100876KY100878KY100874KY100872
CBS 140554House dustMT018036MT005561MN943664MN973462
CBS 140553House dustMT018037MT005562MN943665MN973463
CBS 140556House dustMT018038MT005563MN943666MN973464*S. ajacis* *CBS 177.93; PD 90/115*Delphinium* sp.KT389673GU237673GU238168GU237791*S. ajacis*CBS 176.93; PD 86/547*Delphinium* sp.MT018035GU237672GU238167GU237790*S. andigena* *CBS 269.80; PD 75/914*Solanum* sp.MT018026GU237675GU238170GU237817*S. andigena*CBS 101.80; IMI 386090; PD 75/909*Solanum* sp.—GU237674GU238169GU237714*S. artemisiicola*CBS 102636; PD 73/1409*Artemisia dracunculus*KT389674GU237676GU238171GU237728*S. astragali*CBS 178.25; MUCL 9915*Astragalus* sp.MT018030GU237677GU238172GU237792*S. bomiensis* *CGMCC 3.18366; LC 8167*Boraginaceae*KY742189KY742365KY742277KY742123*S. bomiensis*LC 8168*Boraginaceae*KY742190KY742366KY742278KY742124*S. caricae*CBS 119735*Caricae papaya*MN983680MN984054MN973431MN973042*S. caricae*CBS 120720*Sechium edule*MN983681MN984055MN973432MN973043*S. chrysanthemi*CBS 137.96; PD 84/75*Chrysanthemum indicum*MT018011GU237696GU238191GU237783*S. chrysanthemi*CBS 124241; PD 89/1016-4*Chrysanthemum sinense*MT018010MT005550MN943653MN973451*S. chrysanthemi*CBS 500.63; MUCL 8090*Chrysanthemum indicum*MT018012GU237695GU238190GU237871*S. citrulli* *FLAS-F-58996; C5-5*Citrullus lanatus*—KJ855602—KJ855546*S. cucurbitacearum*CBS 214.65; BBA 9963*Cucumis sativus*MT018020MT005553MN943656MN973454*S. crystalliniformis*CBS 771.85; IMI 386091; PD 85/772*Solanum tuberosum*—GU237684GU238179GU237906*S. crystalliniformis* *CBS 713.85; ATCC 76027; PD 83/826*Lycopersicon esculentum*KT389675GU237683GU238178GU237903*S. cucumeris*CBS 386.65*Cucumis sativus*MT018021MT005554MN943657MN973455*S. cucurbitacearum*CBS 233.52—MT018024MT005555MN943658MN973456*S. cucurbitacearum*CBS 133.96; PD 79/127*Cucurbita* sp.KT389676GU237686GU238181GU237780*S. cucurbitacearum*CBS 109171; PD 91/310*Cucurbita* sp.MN983682GU237685GU238180GU237922*S. dennisii* *CBS 631.68; PD 68/147*Solidago floribunda*KT389677GU237687GU238182GU237899*S. dennisii*CBS 135.96; PD 95/4756*Solidago canadensis*MT018019GU237688GU238183GU237782*S. dorenboschii* *CBS 426.90; IMI 386093; PD 86/551*Physostegia virginiana*KT389678GU237690GU238185GU237862*S. dorenboschii*CBS 320.90; PD 86/932*Physostegia virginiana*MT018039GU237689GU238184GU237830*S. helianthi*CBS 155.90*Helianthus annuus*MT018025MT005556MN943659MN973457*S. helianthi* *CBS 200.87*Helianthus annuus*KT389683KT389848KT389761KT389545*S. heliopsidis*CBS 109182; PD 74/231*Heliopsis patula*KT389679GU237691GU238186GU237747*S. hortensis*CBS 572.85; PD 79/269*Phaseolus vulgaris*KT389681GU237704GU238199GU237893*S. hortensis*CBS 104.42—KT389680GU237703GU238198GU237730*S. hortensis*CBS 130.96*Phaseolus vulgaris*MT018027MT005557MN943660MN973458*S. inoxydabilis* *CBS 425.90; PD 81/520*Chrysanthemum parthenii*KT389682GU237693GU238188GU237861*S. loticola* *CBS 562.81; PDDCC 6884*Lotus pedunculatus*KT389684GU237697GU238192GU237890*S. loticola* *CBS 563.81; PDDCC 6799*Lotus pedunculatus*MT018040MT005564MN943667MN973465*S. lupini*CBS 375.84; PD 80/1250*Lupinus mutabilis*MT018028GU237700GU238195GU237844*S. lupini* *CBS 101494; PD 98/5247*Lupinus albus*KT389685GU237699GU238194GU237724*S. nemophilae*CBS 249.38*Nemophila insignis*MT018032MT005560MN943663MN973461*S. nemophilae* *CBS 715.85; PD 74/364*Nemophila insignis*MT018031MT005559MN943662MN973460*S. oculo-hominis* *CBS 634.92; IMI 193307*Corneal ulcer*KT389686GU237701GU238196GU237901*S. papillata*LC 8170*Rumex nepalensis*KY742192KY742368KY742280KY742126*S. papillata* *CGMCC 3.18367; LC 8169*Rumex nepalensis*KY742191KY742367KY742279KY742125*S. papillata*LC 8171*Boraginaceae*KY742193KY742369KY742281KY742127*S. pini* *MFLUCC 18-1549*Pinus* sp.MK434860MK412886MK348019MK347800***S. pogostemonis* *****ZHKUCC 21-0001*****Pogostemon cablin*****MZ203135****MZ203132****MZ191532****MZ156571**
**ZHKUCC 21-0002*****Pogostemon cablin*****MZ203136****MZ203133****MZ191533****MZ156572***S. rhizophilae*XDPOP-RS-9*Populus deltoides*MN422105MN422099MN422103MN422101*S. rhizophilae*XDPOP-RS-16B*Populus deltoides*MN422104MN422098MN422102MN422100*S. rudbeckiae*CBS 109180; PD 79/175*Rudbeckia bicolor*MT018015GU237702GU238197GU237745*S. sambucella* *CBS 130003*Sambucus nigra*MT018029MT005558MN943661MN973459*S. stuijvenbergii* *CBS 144953; JW 132011Garden soilMN824475MN824623MN823300MN823449*S. tanaceti*CBS 131485*Tanacetum cinerariifolium*MT018014MT005551MN943654MN973452*S. trachelii*CBS 379.91; PD 77/675*Campanula isophylla*KT389687GU237678GU238173GU237850*S. trachelii*CBS 384.68*Campanula isophylla*MT018016GU237679GU238174GU237856*S. trachelii*CBS 123.61*Campanula isophylla*MT018017MT005552MN943655MN973453*S. valerianellae*CBS 273.92; PD 82/43*Valerianella locusta*MT018033GU237705GU238200GU237819*S. weymaniae*CBS 144959; JW 201003Garden soilMN824479MN824627MN823304MN823453*Allophoma piperis*CBS 268.93; PD 88/720*Peperomia pereskifolia*KT389554GU237644GU238129GU237816*Al. piperis*PD 90/2011*Peperomia* sp.MT018045GU237645GU238130GU237921^1^ ATCC: American Type Culture Collection, Virginia, USA; CBS, Centraalbureau voor Schimmelcultures (Netherlands); CGMCC, China General Microbial Culture Collection Center (China); MFLUCC, Mae Fah Luang University Culture Collection (Thailand); IMI: International Mycological Institute, JW: Johanna Westerdijk working collection housed at the Westerdijk Fungal Biodiversity Institute, Utrecht, The Netherlands; LC: LeiCai, corresponding author’s personal collection deposited in laboratory, housed at CAS, China; MUCL: Mycotheque de l’Universite catholique de Louvain, Louvain-la-Neuve, Belgium; PD: Plant Protection Service, Wageningen, the Netherlands; PDDCC: Plant Diseases Division Culture Collection, Auckland, New Zealand; UPSC: Uppsala University Culture Collection, Sweden; ZHKUCC, University of Agriculture and Engineering Culture Collection(China). Sequences produced in this study are shown in bold. * ex-type or ex-epitype culture. ^2^ Sequences data; et al. were downloaded from NCBI following Hou et al. [24] and Wei et al. [25].


### 2.3. Disease Symptoms and Pathogenicity Tests

Two isolates obtained from leaf spot tissue samples were used for pathogenicity analysis. Both mycelial plug method and mycelial suspension method were employed. In the pathogenicity test, inoculated leaf tissues began to show necrosis regions after three days of the inoculation. Initial symptoms were small, oval or circular-shaped light brown spots that were gradually expanded (Figure 4). None of the control plants showed any disease symptom (Figure 4). The disease symptoms appeared on the host by using the mycelial suspension method later than that of mycelium plug inoculation. However, in the mycelial suspension the infected area was larger and there were more irregular small spots with the mycelial suspension method. Some leaves with severe disease even shriveled and dropped off after seven days under the high humidity condition. The stem first turned brown on the surface after seven days of incubation and then became dry and withered. When the stems were cut open with a longitudinal section, browning could be observed inside after one month (Figure 4) of inoculation. The pathogen was re-isolated from the diseased leaves, and the consistent fungi were re-isolated from the diseased leaves but not isolated from the control plants. The culture characteristics of the isolated strains were consistent with the inoculated strain, thus fulfilling Koch’s postulates. 

## 3. Discussion

Didymellaceae, which belongs to Pleosporales, is one of the largest families in the fungal kingdom [26]. They include plant pathogens, opportunists, endophytes, and saprobes from a wide range of host [27]. *Stagonosporopsis*, one of the genera in Didymellaceae, can cause many important plant diseases [28]. *Stagonosporopsis caricae* was reported causing leaf spots on the non-conventional fruit crop, *Vasconcellea monoica* (*Caricaceae*) in Brazil [29]. *Stagonosporopsis chrysanthemi* caused ray blight of chrysanthemum and pyrethrum and is present worldwide [30]. *Stagonosporopsis tanaceti* caused ray blight of pyrethrum (*Tanacetum cinerariifolium*), a perennial herbaceous plant cultivated for the extraction of insecticidal pyrethrins in Australia [31]. *Stagonosporopsis vannaccii* had been reported as a plant pathogenic fungus from Brazil, causing anthracnose symptoms on pods of soybean [32]. In China, *Stagonosporopsis vannaccii* caused leaf spot on *Crassocephalum crepidioides* plants in the kudzu (*Pueraria lobata*) garden in Guangxi [33]. *Stagonosporopsis cucurbitacearum* is the main cause of pumpkin gummy stem blight (GSB), one of the most devastating pumpkin crop diseases in north-east China [34].

*Stagonosporopsis* encompasses a wide range of hosts and occurs worldwide [35]. *Stagonosporopsis citrulli* was demonstrated as a pathogen to appear on the 14 species of Cucurbitaceae in the USA [36]. It was also detected in pyrethrum seed and seedlings [37]. The Cucurbitaceae species, such as cucumber, melon, and pumpkin, contained important pathogens, and the fruit rot diseases caused by *S. cucurbitacearum* became a major disease in many parts of the world [38]. In China, *S. cucurbitacearum* has been reported as parasitic on water spinach (*Ipomoea aquatica*) and caused spot blight disease [39]. *Stagonosporopsis oculihominis* was isolated from *Dendrobium huoshanense* as an endophytic fungus [40]. *Pogostemon cablin* leaf spot disease was frequently observed and seriously influenced by the disease in Zhanjiang city, Guangdong Province, China. The causal organism was identified as a new species; *Stagonosporopsis pogostemonis*. Diseased symptoms were characterized by brown spots from the leaf tips. In some cases, these spots can coalesce and form a big scorch-like spot covering a large portion of the leaves. Moreover, some of them form perforation, wither and drop off. 

In the pathogenicity assays, *Stagonosporopsis pogostemonis* showed symptoms similar to those observed in the field. However, stem blight symptom was only observed in the mycelial suspension method while the mycelial plugs method only developed leaf spots. Therefore, it is necessary to study further to understand the primary inoculation source at the field.

Use of mycelial plug in the pathogenicity assays is always a debatable point among pathologists [41]. Inoculation of a fungus with numerous amounts of growth media might result in weak pathogen to become more aggressive [42]. Moreover, when mycelial plugs are used the inoculum cannot be quantified. In this study we observed that mycelial suspension method took longer time than mycelial plug method to develop symptoms. Thus, it gives us the possibility to understand the latent period of this pathogen with the use of mycelial suspension method. 

Similar to other relevant fields in mycology, it is necessary to identify the pathogenic taxa to the species level [43,44]. Species identification and pathogenicity confirmation are the critical steps to develop effective control measures [44]. Moreover, some species identification should go beyond species level to identify different genotypes responsible for variations in pathogenicity [44]. Therefore, future studies are necessary to collect more samples to identify beyond the species level. This is the first report about *Stagonosporopsis* species causing leaf spot and stem blight in *P*. *cablin* worldwide. The pathogenicity of this species was confirmed with Koch’s postulates.

## 4. Materials and Methods

### 4.1. Sample Collection

Ten *Pogostemon cablin* plants with necrotic leaf spots were collected from the field in Zhanjiang City, Guangdong Province, China (E 110°3′, N 21°2′) from the spring to summer in 2020 (even though the disease incidence occurred in 2019, collections were done only in 2020). Photographs (Nikon D300s, Tokyo, Japan) were taken to record the field symptoms on-site, and samples were placed in sterile, transparent zip-lock bags before being taken back to the laboratory for further studies. Relevant information on the sampling time, latitude, and longitude of the sampling sites and plant species were recorded at the time of sample collection.

### 4.2. Fungal Isolation and Purification

The collected samples were washed with running tap water first. Then diseased leaves were cut into small pieces of tissue (approximately 0.5 × 0.5 cm^2^) by using sterile knife at the interface between healthy and diseased leaves. The pathogen was isolated by the tissue isolation method. Tissue samples were first surface sterilized by soaking in 75% ethanol solution for 10 seconds, followed by immersing in 2.5% NaClO solution for 15 seconds, then rinsed with sterile water three times, and finally dried on sterile filter paper [24]. The surface-sterilized tissue samples were placed on potato dextrose agar (PDA) medium containing streptomycin sulphate (100 μg/mL) and incubated at 25 °C until white mycelia were observed around leaf tissue samples. Pure cultures were obtained after three times hyphal tip isolations. The cultures and the herbarium specimens were preserved in the culture collection and herbarium of Zhongkai University of Agriculture and Engineering (ZHKUCC and ZHKU).

### 4.3. DNA Extraction, PCR Amplification and Sequencings

Total genomic DNA (gDNA) of two strains obtained were extracted from fresh fungal mycelia grown on PDA at 25 °C for seven days. Total DNA was extracted using a modified cetyltrimethylammonium bromide (CTAB) method [25]. Molecular identification of the fungal cultures was determined by using LSU [45,46], ITS [47], *β-tubulin* [48], and *rpb2* [49] genes.

A total of 25 μL PCR reaction mixture contained 1 µL of genomic DNA template, 1 µL of each forward and reverse primer (10 µM), 12.5 µL of I-5™ 2× Easy *Taq* PCR SuperMix(+dye) (TransGen Biotech, Beijing, China) and 9.5 µL of deionized distilled water (ddH_2_O). The thermal cycler conditions used in PCR amplification for all gene regions are given in Table 2. The PCR amplified gene regions were sequenced by Guangzhou Tianyi Science and Technology Co. Ltd. (Guangzhou, China).

### 4.4. Phylogenetic Analysis

The LSU, ITS, *rpb2*, and *β-tubulin* sequences were blasted in NCBI BLASTn (https://blast.ncbi.nlm.nih.gov/Blast.cgi). According to BLAST results of LSU, ITS, *rpb2*, and *β-tubulin* sequences, the isolates obtained in this study were closely related to the species in *Stagonosporopsis.* Relevant sequence data were downloaded from NCBI following Hou et al. [28] and Wei et al. [50]. Individual sequence datasets were aligned using MAFFT version 7 at the web server (http://mafft.cbrc.jp/alignment/server, May 2021) [51] and improved manually where necessary using BioEdit [52] (http://www.mbio.ncsu.edu/BioEdit/page2.html). Then, the aligned datasets were concatenated by PhyloSuite version 7. All sequences obtained in this study were submitted to GenBank (Table 1). Phylogenetic analyses were conducted by maximum likelihood (ML) in RAxML [53], maximum parsimony (MP) in PAUP (v4.0) [54], and Bayesian analyses (BI) in MrBayes (v. 3.0b4) [55].

For the MP analysis, ambiguous regions in the alignment were excluded and gaps were treated as missing data. To evaluate tree stability, 1000 bootstrap replications were done. Zero-length branches were collapsed, and all parsimonious trees were saved. Tree parameters; tree-length (TL), consistency index (CI), retention index (RI), relative consistency index (RC), and homoplasy index (HI) were calculated. Kishino-Hasegawa tests (KHT) were conducted to evaluate differences between the trees inferred under different optimality criteria [56]. MrModeltest v. 2.3 [57] was used to identify the evolutionary models for each locus that were used in the Bayesian analysis. The maximum likelihood analyses were conducted using RAxML-HPC2 on XSEDE (8.2.8) [58] in the CIPRES Science Gateway platform [59]. The GTR + I + G evolutionary model was employed with 1000 non-parametric bootstrapping iterations. Bayesian analysis was performed in MrBayes v. 3.0b4 [55]. Six simultaneous Markov chains were run for 106 generations, sampling the trees at every 200th generation. From the 10,000 trees obtained, the first 2000 representing the burn-in phase were discarded. The remaining 8000 trees were used to calculate posterior probabilities in a majority rule consensus tree. The constructed phylogenetic tree was visualized in FigTree v1.4.2 and edited by Adobe Illustrator CS6. The final sequence alignment generated in this study was submitted to TreeBASE (https://treebase.org/treebase-web/home.html) under submission ID 28457.

### 4.5. Morphological Identification

The morphological characterizations of the fungal isolates were carried out based on comprehensive observation of colony characters and microscopic morphology of strain ZHKUCC 21-0001. Culture characteristics were recorded from colonies grown on PDA, malt extract agar (MEA), and oatmeal agar (OA) plates at 25 °C in 12 h dark/12 h light photoperiod for 15–30 days until sporulation [60]. Colony diameters were measured by the crisscross method on the fifth day and growth rate was calculated. The phenotypic characteristics such as colony shape, size, color, exudates, and colony margins were observed, recorded and photographed. Pycnidia were cut into 30 µm thin sections by a freezing sliding microtome (Bio-Key science and technology Co., LTD., LEICA CM1860, Weztlar, Germany) for photographing and measuring. Microscopic characters were observed and photographed using Nikon ECLIPSE 80i microscope (Nikon, Tokyo, Japan) and measurements were taken using NIS-Elements BR 3.2. Measurements of spore length and width of 50 spores were taken. The mean values and standard deviation were calculated with Microsoft Excel.

### 4.6. Pathogenicity Test

Pathogenicity tests were conducted using *Pogostemon cablin* healthy potted seedlings. Inoculations were done by both mycelial plug method and mycelial suspension inoculation. These assays were done as non-wounded leaves and a small wounded by sterilized needle of stems. Healthy leaves of the same developmental stages were selected, and then the leaves and stems were surface sterilized with 75% alcohol. Fresh wounds were made with a sterilized needle. 6 mm-diameter mycelial plugs were put in the leaf and were inoculated on the surface of healthy young leaves. The 10% mycelial suspension (10 mg [wet weight]/100 ml [volume]) were crushed using a juice extractor (MJ-BL25B2, Guangdong Midea Household Appliance Manufacturing Co., Ltd., Guangdong, China), and wiped on the leaves and stems by sterile cotton. The leaves and stems were then covered with wet cotton and sealed with Parafilm and bagged for moisturizing. Leaves inoculated with water were used as the control. Inoculated plants were kept on the shelf in the greenhouse (25 °C) with artificial lighting (14-h period of supplementary lighting/10-h dark) each day. Disease symptoms were checked daily for 2–7 days. Once the disease was developed, the pathogen was re-isolated to confirm Koch’s postulates.

## Figures and Tables

**Figure 1 pathogens-10-01093-f001:**
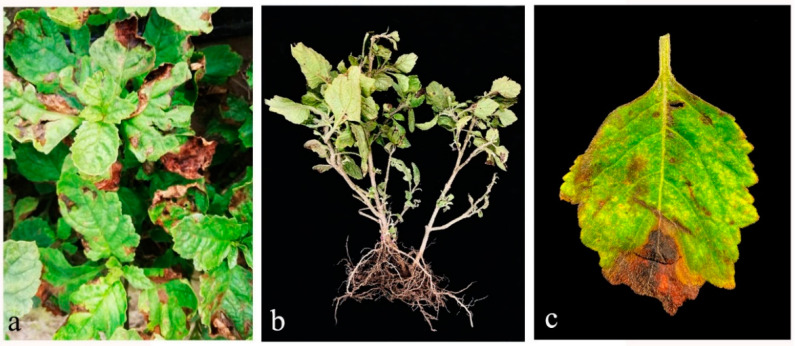
Field symptoms of leaf spot and stem blight caused by *Stagonosporopsis pogostemonis* (**a**) infected plants in the field with leaf spot starting from the tips of leaves. (**b**) Infected plant with wilting appearance. (**c**) Infected leaf with characteristic lesion, which starts with foliar marginal brown discoloration, then spreading towards the inside leaf lamina.

**Figure 2 pathogens-10-01093-f002:**
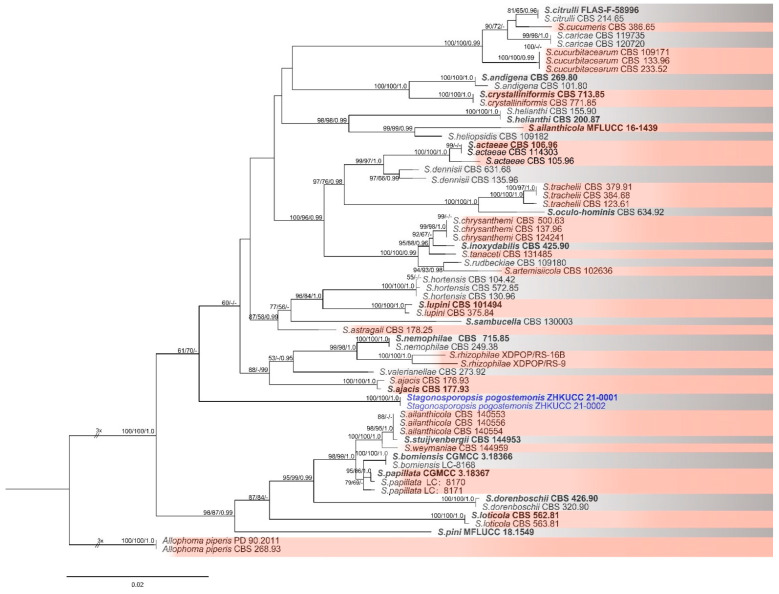
The best scoring RAxML tree obtained using the combined dataset of LSU, ITS, *rpb2*, and *β-tubulin* sequences. *Allophoma piperis* (CBS 268.93 and PD 90/2011) was used to root the tree. Bootstrap support values equal to or greater than 50% in ML and MP and BYPP equal or greater than 0.95 are shown as ML/MP/BYPP above the respective node. The isolates belonging to the current study are given in blue. Ex-type strains are bold. Expected number of nucleotide substitutions per site is represented by the scale bar.

**Figure 3 pathogens-10-01093-f003:**
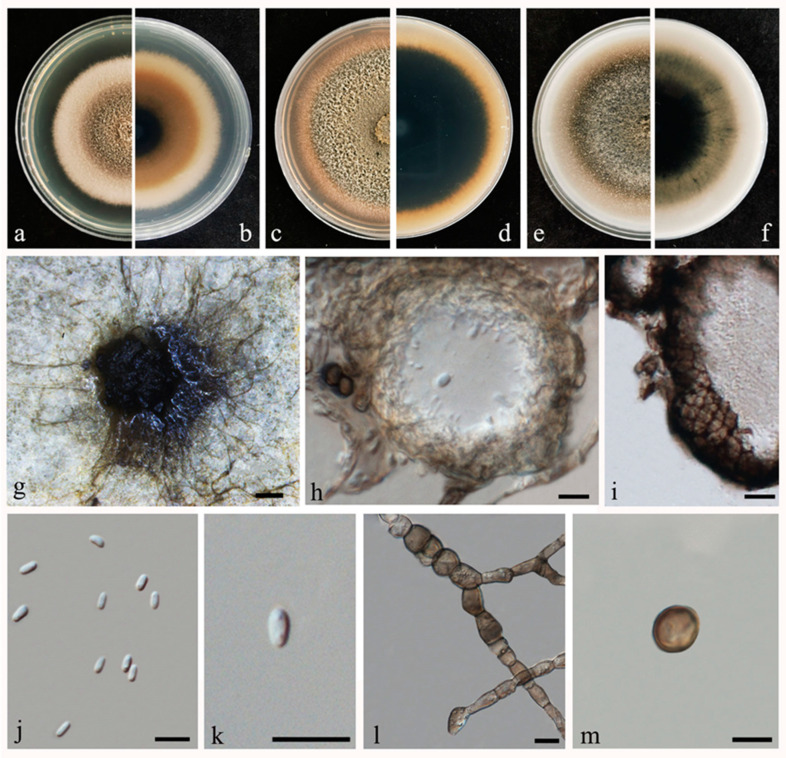
Morphological characteristics of *Stagonosporopsis pogostemonis* (Ex holotype ZHKUCC 21-0001). (**a**,**b**) Front and reverse view on PDA after five days at 28 °C. (**c**,**d**) Front and reverse view on MEA after seven days at 28 °C. (**e**,**f**) Front and reverse view on OA after seven days at 28 °C. (**g**) Pycnidia on PDA. (**h**) Longitudinal section of pycnidia on PDA. (**i**) Pycnidia wall. (**j**,**k**) Conidia. (**l**) Developing chlamydospores. (**m**) Chlamydospore. Scale bars: g = 200 μm; h, k, l, and m = 10 μm.

**Figure 4 pathogens-10-01093-f004:**
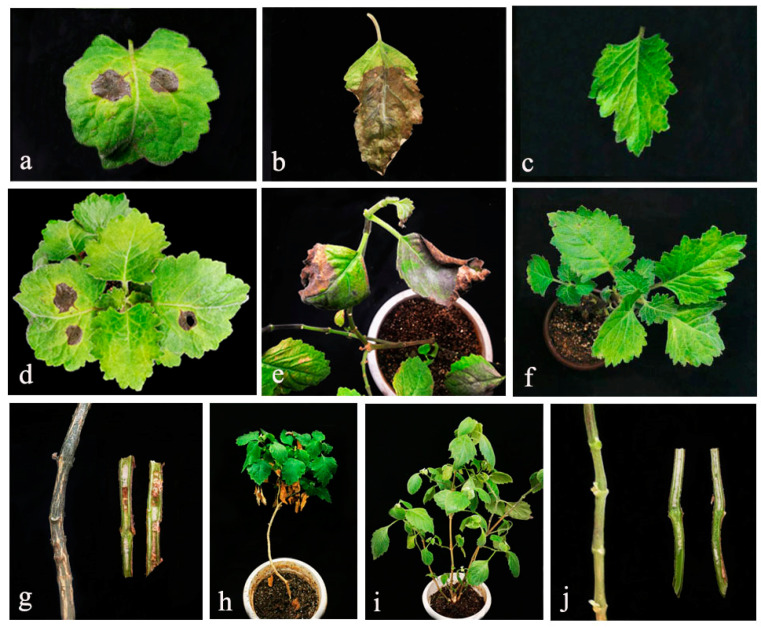
Pathogenicity tests results of *Stagonosporopsis pogostemonis* inoculated into potted *Pogostemon cablin* plants. Infected leaf at seven days post inoculation with (**a**) mycelial plug and (**b**) mycelial suspension. (**c**) Control leaf. Infected plant at seven days post inoculation with (**d**) mycelial plug and (**e**) mycelial suspension. (**f**) Control plant. Vascular tissues and stem of plants inoculated with mycelial suspension (**g**,**h**) one month post inoculation and (**i**,**j**) control plants.

**Table 2 pathogens-10-01093-t002:** Gene regions and respective primer pairs used in the study.

Amplified Gene.	Primer Pairs	Optimized PCR Protocols	References
*rpb2*	fRPB2-5F	95 °C: 5 min, (94 °C: 30 s, 54 °C: 30 s, 72 °C: 1 min) × 32 cycles, 72 °C: 10 min	[49]
	fRPB2-7cR		
*β-tubulin*	Bt2a	95 °C: 5 min, (94 °C: 30 s, 58 °C: 30 s, 72 °C: 1 min) × 32 cycles, 72 °C: 10 min	[48]
	Bt2b		
LSU	LROR	95 °C: 5 min, (94 °C: 30 s, 49 °C: 30 s, 72 °C: 1 min) × 32 cycles, 72 °C: 10 min	[45,46]
	LR5		
ITS	ITS1	95 °C: 5 min, (94 °C: 30 s, 53 °C: 30 s, 72 °C: 1 min) × 32 cycles, 72 °C: 10 min	[47]
	ITS4		

## Data Availability

The sequence data generated in this study are deposited in NCBI GenBank (https://www.ncbi.nlm.nih.gov/genbank). All accession numbers are given in Table 1. The Final alignment generated in this study available in TreeBASE under the accession number 28457.

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
