# Peer review of "Stagonosporopsis pogostemonis: A Novel Ascomycete Fungus Causing Leaf Spot and Stem Blight on Pogostemon cablin (Lamiaceae) in South China"

_pathogens, 2021, doi:10.3390/pathogens10091093_

Round 1
Reviewer 1 Report
In this manuscript the authors describe the identification and characterisation of a novel ascomycetes fungus, Stagonosporopsis pogostemonis, from Pogostemon cablin plants showing leaf spot and stem blight symptoms. The authors have characterised the new fungus through its morphological features and with molecular based approaches that are well supported by the extended phylogenetic analysis. Finally, through their pathogenicity assays have fulfilled Koch's postulates and this is the first report of the fungus infecting P. cablin worldwide.
However, the authors should include more information about the number of the samples analysed and sequenced. My specific comments and suggestions, as well as corrections on the language of the manuscript, are included in the attached file.

Author Response
Dear Reviewer
Thank you very much for your extensive and insightful reviewer comments. We have rewritten sections of the manuscript and the figures are modified. We will respond to the comments below.
In addition, all the other comments were corrected and revised parts were highlighted. We hope that these revisions will answer all the questions raised by the reviewers and facilitate them in reviewing the manuscript.

Reviewer 2 Report
Dear Authors, please, check Itroduction section and correct titles of Figures 1 and 4 (they should include name of the fungus causing disease symtopms,).
In the case of Stagonosporopsis fungi it is always interesting to study herbarium material because conidial morphology in vivo may differ from conidia produced in in vitro. Do you have this data?
Please, indicate more accurately conditions for artificial inoculation: inoculum concentration, wetness period, incubation conditions etc. Describe size of lesions. Did you re-isolate the fungus from stems. You wrote on leaf wounding, was it effective for disease stimulation? In results you discussed inoculation with mycelial suspension while conidial suspension was used in methods.
Fig. 3K - is not good to illustarte chlamidospores, please, change it for more complex aggregations of melanized cells.
Author Response
Dear Reviewer
Thank you very much for your extensive and insightful reviewer comments. We have rewritten sections of the manuscript and the figures are modified. We will respond to the comments below.
- Dear Authors, please, check the Introduction section and correct titles of Figures 1 and 4 (they should include the name of the fungus causing disease symptoms,).
Thank you very much for this comment. We have changed the figure titles as follows. Hope it will give a better resolution. “Figure 1. Field symptoms of leaf spot and Stem Blight caused by Stagonosporopsis pogostemonis. Figure 4. Pathogenicity results of Stagonosporopsis pogostemonis inoculated into potted Pogostemon cablin plants”
- In the case of Stagonosporopsis fungi, it is always interesting to study herbarium material because conidial morphology in vivo may differ from conidia produced in vitro. Do you have this data?
Thank you very much for mentioning this. We did not have any data about in vivo sporulation. We directly isolated from diseased samples rather than looking for fruiting bodies in the field condition.
- Please, indicate more accurately conditions for artificial inoculation: inoculum concentration, wetness period, incubation conditions etc. Describe size of lesions. Did you re-isolate the fungus from stems. You wrote on leaf wounding, was it effective for disease stimulation? In the results you discussed inoculation with mycelial suspension while the conidial suspension was used in methods.
Thank you very much for the comment. We would like to provide answers point by point as below.
Inoculation conditions: We have mentioned the inoculum conditions in detail in the materials and methods section. The concentration; 10% mycelial suspensions (10 mg [wet weight] / 100 ml [volume]) The condition; Inoculated plants were kept on the shelf in the greenhouse (25 °C) with artificial lighting (14-h period of supplementary lighting / 10-h dark) each day. Disease symptoms were checked daily for 2–7 days. Therefore, we did not add these details to the Results section.
Size of lesions: We did not measure the size of lesions, but we compare the sizes in both methods.
Re-isolate the fungus from stems: Pathogen was reisolated from leaves but not from the stem.
Effectiveness of the wounding: in the inoculation essays wounded leaves developed the symptoms earlier. Thus, we also think wounded inoculation is an effective method for disease stimulation.
Mycelial suspension: we are sorry for the mistake done here. It should be mycelial suspension, not spore suspension. We have corrected this throughout the manuscript. Thank you very much for pointing this.
We hope all these explanations are acceptable.
- 3K - is not good to illustrate chlamydospores, please, change it for more complex aggregations of melanized cells.
Thank you very much for the comment. We have modified figure 3 accordingly. We hope the changes are acceptable.
In addition, all the other comments were corrected and revised parts were highlighted. We hope that these revisions will answer all the questions raised by the reviewers and facilitate them in reviewing the manuscript.

Round 2
Reviewer 2 Report
See comments in the attaced file.

Author Response
Dear Reviewer
Thank you very much for your reviewer comments. We have corrected the minor comments from you and all changes are highlighted. However, we would like to have one more clarification.
According to your comment in Figure 4. The figure title should be “Disease symptoms on Pogostemon cablin 7 days post artificial inoculation with mycelia of Stagonosporopsis pogostemonis” However we have a small concern, whether this might be affected to the reader to get the idea of this figure is about disease symptoms on the field. Could you please let us know will you agree with our point and keep the figure title as “Figure 4. Pathogenicity tests results of Stagonosporopsis pogostemonis inoculated into potted Pogostemon cablin plants”.
In addition, all the other comments were corrected, and revised parts were highlighted.